# Galvanic Corrosion of SA106 Gr.B Coupled with Magnetite in Alkaline Solution at Various Temperatures

**DOI:** 10.3390/ma12040628

**Published:** 2019-02-20

**Authors:** Soon-Hyeok Jeon, Geun Dong Song, Sang Ji Kim, Do Haeng Hur

**Affiliations:** Korea Atomic Energy Research Institute, 989-111 Deadeok-daero, Yuseong-gu, Daejeon 34057, Korea; junsoon@kaeri.re.kr (S.-H.J.); sgd84@kaeri.re.kr (G.D.S.); sjkim3@kaeri.re.kr (S.J.K.)

**Keywords:** carbon-manganese steel, magnetite, polarization, galvanic corrosion, temperature

## Abstract

The effect of temperature on the galvanic corrosion behavior of SA106 Gr.B carbon-manganese steel was studied in an alkaline aqueous solution at various temperatures (30, 60, and 90 °C) via electrochemical corrosion tests. At all temperatures studied, carbon-manganese steel acted as the anode of the galvanic cell composed of carbon-manganese steel and magnetite because the corrosion potential of carbon-manganese steel was significantly lower than that of magnetite. The corrosion current density of carbon-manganese steel significantly increased due to the galvanic effect irrespective of temperature used in this study. With the increase in temperature, the extent of the galvanic effect on the corrosion current density of carbon-manganese steel and reductive dissolution of magnetite gradually increased. When the area ratio of magnetite to carbon-manganese steel increased, the corrosion rate of the carbon-manganese steel in contact with magnetite further increased.

## 1. Introduction

SA106 Gr.B, carbon-manganese steel, is widely used for high-temperature services such as feed-water pipes in pressurized water reactors (PWRs) [1], drain lines in boiling water reactors (BWRs) [2], and high-pressure ammonia feed lines from the ammonia heater to the urea reactor in fertilizer plants [3]. This material is susceptible to pitting corrosion [3] and flow accelerated corrosion (FAC) [4] due to its low corrosion resistance. 

In the secondary system of PWRs, magnetite is the major corrosion component of oxide layers on the surface of SA106 Gr.B piping [5,6,7]. Magnetite particles are moved with the feedwater into the secondary side of a steam generator (SG) and deposited onto the surfaces of SG tubes and SG structural components [5,6,7]. The deposition of magnetite decreases the heat transfer efficiency of the SG tubes and leads to blockage of the waterway [8,9]. Additionally, aggressive chemical impurities, such as chloride ion (Cl^−^), various sulfur species (S^2−^, S_2_O_3_^2−^, SO_3_^2−^ and SO_4_^2−^), metallic lead (Pb), and copper (Cu) particles, are concentrated within the micro-pores existed in the magnetite deposits by local boiling [10,11]. The aforementioned phenomenon increases the corrosion rate of SG structural materials [10,11].

Magnetite deposits are electrically connected with various secondary system materials that are in contact with magnetite in nuclear power plants. Magnetite deposits exhibit a porous morphology [12,13], a high electrical conductivity (approximately 10^2~3^/Ω·cm at room temperature) [14], and a relatively small energy band gap (0.1 eV) [15], thereby indicating that magnetite acts in a manner similar to a metal. Therefore, magnetite can be expected to affect the corrosion behavior of secondary materials when it comes into contact with the materials in the secondary water systems of PWRs. 

Thus, several studies on the galvanic effect of magnetite on the corrosion behavior of various secondary materials and water chemistry conditions were recently reported [16,17,18,19,20,21,22]. With respect to SG tube materials, such as Alloy 600 and Alloy 690, the corrosion rate of SG tube materials significantly increased due to a shift in their corrosion potentials in the positive direction in alkaline aqueous solutions [16,17,18,19]. Additionally, magnetite exerts a synergistic effect on increase in the corrosion rate of SG tube materials with chemical impurities, such as chloride ions [18] and lead ions [19]. The effect of magnetite contact on the galvanic corrosion of SA508 SG tubesheet material was also investigated in ethylenediaminetetraacetic acid (EDTA)-based chemical cleaning solutions through electrochemical and immersion tests [20]. SA106 Gr.B carbon-manganese steel corresponded to the anode when this steel and magnetite were electrically connected in an alkaline solution at room temperature [21]. Accordingly, when the magnetite and SA106 Gr.B are galvanically contacted to the equivalent area ratio, the corrosion rate of the coupled steel was increased by approximately 3.5 times than that of the non-coupled steel [21].

However, the extent of the galvanic effect between carbon-manganese steel and magnetite decreased with the addition of 100 ppm polyacrylic acid (PAA) [22]. Based on this research, the corrosion mechanism of secondary system materials in contact with magnetite has been adequately elucidated and the galvanic effect has also been reflected on the actual nuclear power plants. 

However, there is a no investigations on the effect of various temperatures on the galvanic corrosion behavior of SA106 Gr.B, which is widely used as a piping material for the secondary systems of PWRs. Specifically, during the shutdown period of PWRs, SGs are typically into wet layup to protect the internal parts from corrosion and to promote the return of chemical impurities from deposits and crevices [23]. Wet layup condition is in operating chemistry at low temperature below 93 °C without exposure to air [23]. However, the effect of temperature on the galvanic corrosion behavior of SA106 Gr.B due to magnetite has not been studied in wet layup conditions. 

Additionally, the combined effect of temperature and area ratio (AR) on the degree of the galvanic effect of SA106 Gr.B has not been evaluated to date. In real situations of nuclear power plants, a small surface area of SA106 Gr.B piping is exposed to secondary coolant water within micro-pores at the interface between the porous magnetite and secondary materials. In this situation, the small exposed area of carbon-manganese steel to the water is in electrical contact with a relatively large surface area of magnetite. Furthermore, galvanic corrosion between magnetite and SA106 Gr.B piping can also occur due to spallation and exfoliation, which mechanically removes corrosion products formed on the carbon-manganese steel [24,25]. Hence, it is crucial to evaluate the effect of temperature and AR on the degree of galvanic corrosion between the magnetite and SA106 Gr.B carbon-manganese steel. In addition, in a previous study on SA106 Gr.B [21], the predicted values were only shown using potentiodynamic polarization curves. Therefore, it is necessary to verify the real galvanic behavior through the zero resistance ammeter (ZRA) measurements.

Thus, the objective of this study involves investigating the effects of temperature and AR on the galvanic corrosion behavior of SA106 Gr.B carbon-manganese steel at various temperatures in a wet layup condition of the secondary water system of a PWR. In order to elucidate the corrosion behavior of pure magnetite and its effects, magnetite layers were prepared via the electrodeposition method. Galvanic corrosion behavior of SA106 Gr.B in contact with magnetite was investigated via various electrochemical corrosion tests. The corrosion rate of SA106 Gr.B by galvanic coupling with magnetite was predicted by polarization curves, and verified by ZRA measurements. The combined effect of AR and temperature on the extent of the galvanic effect was also predicted and discussed.

## 2. Experimental Methods 

### 2.1. Specimen Preparation and Test Solution

SA106 Gr.B pipe material was cut into specimens with a size of 10 × 5 × 1 mm for the electrochemical corrosion tests. The specimens were sequentially ground by using silicon carbide papers down to 1000-grit and subsequently cleaned in acetone by using an ultrasonic cleaner (JAC-3010, KODO, Hwaseong, Korea). After the specimens were cleaned, they were directly dried to remove the remaining moisture in a vacuum oven (HB-501S, Hanbaek Scientific Co., Bucheon, Korea) for 1 h at approximately 60 °C. During the test preparation, the samples were kept in a vacuum desiccator (V-002, ThreeShine Co., Daejeon, Korea) under a vacuum degree of 0.09 MPa and 25 °C to prevent oxidation reaction in the air. The chemical composition of SA106 Gr.B is shown in Table 1. 

### 2.2. Preparation of Working Electrodes for the Electrochemical Tests

In the study, each working electrode (SA106 Gr.B and magnetite) was required to elucidate the electrochemical corrosion properties of SA106 Gr.B and magnetite. First, in order to produce an SA106 Gr.B working electrode, the specimen was spot-welded to an iron wire by using a thermocouple welder (HotSpot II, DCC Co., NJ, USA). Subsequently, the specimen was placed in a polytetrafluoroethylene (PTFE) tube for electrical insulation from the test solution. A silicon resin was then coated around the weld part to prevent the test solution from permeating into any remaining gaps or crevices. 

A magnetite working electrode was fabricated by the electrodeposition of magnetite layer over the whole surface of the prepared SA106 Gr.B working electrode. The electrodeposition solutions corresponded to a mixture of 0.1 M triethanolamine (TEA, C_6_H_15_NO_3_), 0.043 M ferric sulfate hydrate (Fe_2_(SO_4_)_3_), and 2 M sodium hydroxide (NaOH). The electrodeposition of magnetite was performed in a three-electrode cell by using PAR273 potentiostat equipment (Ametek, Berwyn, PA, USA) and Power Suite software (Ametek, Berwyn, PA, USA). A platinum wire and a saturated calomel electrode (SCE) were used as a counter electrode and a reference electrode, respectively. The magnetite layer was deposited in the mixed deposition solution in a deposition bath at an applied potential of −1.05 V_SCE_ for 30 min at 80 °C. In a previous study, iron-based films were electrodeposited on SA106 Gr.B substrate at various potentials in Fe(III)-TEA solution at 80 °C [26]. The applied potentials greatly affected the morphology, structure, thickness, and surface roughness of the electrodeposited iron-based films [26]. Based on the results, the deposition potential range from −1.05 V_SCE_ to −1.11 V_SCE_ was proper to produce pure, adhesive, and homogeneous magnetite layer [26]. Among these potentials, an applied potential of −1.05 V_SCE_ was selected in this study because the surface morphology of the magnetite deposited at this potential was most similar in shape to the magnetite flakes taken from an operating PWR [27]. Figure 1 shows the schematic of the test apparatus for the electrodeposition. 

Figure 2 shows photographs of a SA106 Gr.B specimen and an electrodeposited magnetite specimen. After the electrodeposition, the deposited magnetite specimen was analyzed via focused ion beam-scanning electron microscopy (FIB-SEM, FEI, Hillsboro, CA, USA) to closely observe the cross-section of the magnetite layer. The working distance was 10 mm and the acceleration voltage was 5 kV. The morphology, chemical composition, and layer thickness of the magnetite were analyzed by using a FIB-SEM attached with an energy dispersive X-ray spectroscope (EDS). An X-ray analysis was conducted to accurately identify the structure of the SA106 Gr.B and electrodeposition layer via an X-ray diffractometer (XRD, Rigaku, Tokyo, Japan) with Cu-Kα radiation (λ = 1.5406 Å).

### 2.3. Electrochemical Corrosion Tests

Two different electrochemical corrosion tests (potentiodynamic polarization test and ZRA measurement) were performed to investigate the effect of temperature on the extent of galvanic corrosion between SA106 Gr.B and magnetite in an alkaline solution. All test solutions were prepared immediately prior to each test from demineralized water. An alkaline solution with a pH value of 9.5 at room temperature was used in the electrochemical tests. The pH of the test solution was controlled via ethanolamine (ETA, C_2_H_7_NO), a pH control agent, which is used as an ammonia substitute in the secondary coolant system of nuclear power plants since 2001 [23]. The electrochemical corrosion tests were performed at three temperatures (30, 60, and 90 °C) to simulate the change in temperature from cold shutdown to the wet layup state [28]. In order to retain the deaerated conditions, high-purity nitrogen gas (purity 99.99%) was continuously input into the test solutions at a flow rate of 100 cm^3^/min during all the electrochemical tests. In order to simulate the real secondary water chemistry of PWRs in wet layup condition, the pH 9.5 at room temperature and deaerated condition was selected in this work.

The potentiodynamic polarization tests were performed in a three-electrode cell via the PAR273 potentiostat (Ametek, Berwyn, PA, USA) and Power Suite software (Ametek, Berwyn, PA, USA). The reference and counter electrodes corresponded to the same materials as mentioned above. The exposed surface of magnetite and SA106 Gr.B specimens was 1.3 cm^2^. The open circuit potential (OCP) reached a stable state after immersion for 1 h. After the OCP was stabilized, the OCP of the two materials was recorded for 1 h. After the OCP was measured, the polarization curve was scanned from the OCP to the negative (cathodic) or positive (anodic) direction with a scan rate of 1 mV/s. The anodic and cathodic polarization curves were combined into a final curve. The corrosion current densities (*i*_corr_) of SA106 Gr.B and magnetite at the corrosion potentials (*E*_corr_) were measured via the Tafel extrapolation method of the cathodic curve.

The real galvanic behavior (galvanic potential and galvanic current density) between SA106 Gr.B and magnetite was also evaluated through the ZRA measurement by using a Gamry Reference 600 instrument potentiostat (Gamry, Warminster, PA, USA). With respect to galvanic coupling measurements, the exposed surface of magnetite and SA106 Gr.B specimens was 1.3 cm^2^. The area ratio between SA106 Gr.B and magnetite specimens was equivalent (1:1). SA106 Gr.B specimen was connected to the working electrode and magnetite specimen was connected to another working electrode. The reference electrode was placed as close as possible between the two coupled electrodes. After the OCP reached the stable state, the two electrodes were electrically connected in the test solution. The galvanic potential (*E*_couple_) and galvanic current density (*i*_couple_) of SA106 Gr.B coupled with magnetite was measured for 3600 s. Two electrochemical corrosion techniques were performed three times per specimen to assess reproducibility, and the results indicated good reproducibility. Figure 3 shows the schematic of the three-electrode cell for the electrochemical corrosion tests.

## 3. Results and Discussion

### 3.1. Electrodeposited Magnetite Layer on SA106 Gr.B Substrate

Figure 4 shows the SEM micrographs of the surface of the magnetite layer deposited on the SA106 Gr.B substrate. As shown in the Figure 4, the magnetite appears to grow in a polyhedral shape. 

The layer thickness was evaluated to be approximately 4 to 6 μm, as indicated by the cross-section image (Figure 5a). The magnetite layer exhibits uniform characteristics. Defects, such as holes and crevices were not observed within the layer or at the interface between the magnetite layer and the SA106 Gr.B substrate, indicating that the layer was tightly connected to the SA106 Gr.B substrate. SEM-EDS analysis was performed in order to obtain the stoichiometric ratio between the Fe and O in the electrodeposited layer. The EDS line analysis of the magnetite is presented in Figure 5b. The Fe/O atomic ratio of the deposited layer is about 0.75~0.78, consistent with the theoretical Fe/O atomic ratio of magnetite. The EDS elemental mapping analysis of the magnetite layer is also shown in Figure 5c,d. The results also indicate that the magnetite layer was composed of O and Fe. In order to ensure that pure magnetite was adequately formed on the substrate, point EDS analysis was also performed at two positions, which were numbered in Figure 5a. The quantitative results in Table 2 indicate that the electrodeposited layer corresponded to pure magnetite because it was composed of approximately 42 to 43 at.% Fe and 56 to 57 at.% O. 

In order to cross-check the structure of the electrodeposited layer, XRD analysis was also performed. Figure 6 shows the XRD patterns of the SA106 Gr.B and electrodeposited magnetite layer on SA106 Gr.B substrate. SA106 Gr.B and magnetite layer were both presented as highly crystalline. SA106 Gr.B only has peaks corresponding to universal XRD data for iron (powder diffraction file (PDF) No. 00-006-0696) because SA106 Gr.B composed of about 98% iron (Figure 6a). As shown in Figure 6b, the position and relative intensity of the diffraction peaks of the layer matched well with the XRD data for magnetite (PDF No. 00-019-0629). The SEM-EDS and XRD results indicate that the electrodeposited layer corresponded to pure magnetite and the magnetite working electrode was appropriate to measure the electrochemical corrosion behavior of pure magnetite electrode itself without exposing the SA106 Gr.B substrate to the test solutions through the magnetite layer.

### 3.2. Electrochemical Corrosion Behavior 

In order to elucidate the effect of temperature on the corrosion behavior between the SA106 Gr.B and magnetite, the *E*_corr_ of the two materials was measured at various temperatures via a potentiostat. Figure 7a shows the *E*_corr_ of the SA106 Gr.B and magnetite in the test solution at 30, 60, and 90 °C with respect to the test time. As the temperature increased from 30 °C to 90 °C, the *E*_corr_ of magnetite continuously decreased from −0.410 V_SCE_ to −0.602 V_SCE_. However, the *E*_corr_ of SA106 Gr.B decreased from −0.780 to −0.855 V_SCE_ as temperature increased from 30 °C to 60 °C and did not significantly change in the range of 60–90 °C. 

As shown in Figure 7b, the *E*_corr_ of SA106 Gr.B was higher than that of magnetite by 264 to 382 mV at all temperatures. Thus, when SA106 Gr.B and magnetite are in electrical contact in the same solution, magnetite acts as the cathode in the galvanic pair, while SA106 Gr.B behaves as the anode. A corrosion potential difference between two materials should exceed about 50 mV to induce an increased galvanic corrosion [29]. Consequently, the corrosion of SA106 Gr.B will be accelerated when SA106 Gr.B and magnetite are galvanically coupled. 

Figure 8 shows the potentiodynamic polarization curves of SA106 Gr.B and magnetite in the test solution at various temperatures. In order to quantitatively measure the effect of the temperature on the galvanic behavior between SA106 Gr.B and magnetite, the *i*_corr_ values of the SA106 Gr.B and magnetite were calculated via cathodic Tafel extrapolation of the polarization curves. The *E*_couple_ and the *i*_couple_ of the galvanic couple between SA106 Gr.B and magnetite were also calculated via mixed potential theory. The electrochemical parameters are listed in Table 3. 

As the temperature increased from 30 °C to 90 °C, the *E*_corr_ and *i*_corr_ of both magnetite and SA106 Gr.B decreased and increased, respectively. The reason for the increase of the *i*_corr_ of the two materials is that the increase in temperature can promote the transport rate of ions and reduce the pH value. When SA106 Gr.B and magnetite are electrically connected in an equivalent area (AR of cathode to anode = 1), it is expected that the corrosion rate of SA106 Gr.B significantly increases due to the galvanic effect at all temperatures. As shown in Table 3, the corrosion rate of SA106 Gr.B will be increased owing to the galvanic effect with magnetite as follows: By about 2.9 times from 2.51 μA/cm^2^ to 7.36 μA/cm^2^ at 30 °C; by about 3.6 times from 3.42 μA/cm^2^ to 12.42 μA/cm^2^ at 60 °C; and by about 2.5 times from 6.91 μA/cm^2^ to 17.13 μA/cm^2^ at 90 °C. 

The oxidation and reduction reactions of magnetite can occur as following reactions [30,31,32,33]:Fe_3_O_4_ + OH^−^ + 4H_2_O → 3Fe(OH)_3_ (s) + e^−^ (oxidation reaction) (1)
Fe_3_O_4_ + OH^−^ + 4H_2_O + 2e^−^ → 3Fe(OH)_3_^−^ (reductive dissolution reaction)(2)

Equations (1) and (2) indicate the oxidation and reduction reactions of magnetite with water and hydroxide ions in the test solution, respectively.

In the case of magnetite coupled with SA106 Gr.B, carbon-manganese steel is observed to aid the magnetite dissolution based on the following reactions (3) and (4) [6,34]: Fe (carbon-manganese steel) → Fe^2+^ + 2e^−^(3)
Fe_3_O_4_ + OH^−^ + 4H_2_O + Fe → 3Fe(OH)_3_^−^ + Fe^2+^(4)

The corrosion of SA106 Gr.B increases by the galvanic couple and thus supplies excess electrons. Therefore, the reductive dissolution reaction rate of magnetite increases with increasing electrons due to the corrosion of SA106 Gr.B. Consequently, it is expected that the galvanic couple between SA106 Gr.B and magnetite accelerates both the corrosion of carbon-manganese steel and the reductive dissolution of magnetite.

Figure 9 shows the actual *E*_galvanic_ and *i*_galvanic_ of the galvanic couple between SA106 Gr.B and magnetite, which were measured by ZRA in the test solution at various temperatures. The data were measured at equal areas of SA106 Gr.B and magnetite specimens. The actual *E*_galvanic_ was shifted in the negative direction with the increase in temperature (Figure 9a). When comparing Figure 9a and Table 3, the actual *E*_galvanic_ almost coincides with the *E*_couple_ calculated from the potentiodynamic curve via mixed potential theory. Meanwhile, in the actual SA106 Gr.B/magnetite galvanic couple, the *i*_galvanic_ of SA106 Gr.B showed a positive current value (Figure 9b), indicating that SA106 Gr.B was the anode of the couple. The actual *i*_galvanic_ changed gradually until it reached a constant value. The actual *i*_galvanic_ also increased with the increase in temperature. 

Although the ZRA technique presented lower corrosion current densities than those calculated by the potentiodynamic curves and mixed potential theory, the results indicated a similar tendency.

As previously mentioned, in the actual situation of the secondary water systems of PWRs, a large surface area of magnetite (large cathode) and a small surface area of carbon-manganese steel (small anode) are observed. In addition to the removal of magnetite deposits through exfoliation or spallation phenomena, a cavitation can occur in severe turbulent flow, which also removes the magnetite layer in localized areas and results in the formation of a galvanic cell with a large cathode and a small anode. Therefore, it is necessary to consider the AR effect on the galvanic extent of SA106 Gr.B. In order to calculate the AR effect on galvanic corrosion, we rearranged the polarization curves of magnetite with AR = 1, 5, 10 and calculated the galvanic current densities via mixed potential theory. Figure 10 shows the effect of the AR on the galvanic behavior between SA106 Gr.B (anodic curves) and magnetite (cathodic curves) in the test solutions at various temperatures. The changes in the *i*_couple_ caused by AR = 1, 5, and 10 are shown in Table 4. The *i*_couple_ significantly increased with the increase of the AR. Especially, the *i*_couple_ significantly increased when the AR changed from 1 to 5. Based on the electrochemical corrosion results, the corrosion rate of SA106 Gr.B significantly increased in a site where both temperature and AR increased.

## 4. Conclusions

The effect of temperature and AR on the galvanic corrosion behavior of SA106 Gr.B carbon-manganese steel was examined in simulated secondary water conditions at various temperatures (30, 60, and 90 °C) via two electrochemical methods. The increased corrosion rate of SA106 Gr.B by galvanic coupling with magnetite was predicted by polarization curves, and verified by ZRA measurements.
Based on the polarization curves and ZRA results, SA106 Gr.B acted as the anode of the galvanic cell between SA106 Gr.B and magnetite because the corrosion potential of SA106 Gr.B was lower than that of magnetite at all temperatures.The corrosion rate of SA106 Gr.B and the reductive dissolution rate magnetite significantly increased due to the galvanic effect irrespective of temperatures.The extent of the galvanic effect on the corrosion rate of SA106 Gr.B and reductive dissolution of magnetite gradually increased with the increase of temperature. Furthermore, the corrosion rate of SA106 Gr.B coupled with magnetite further increased with increasing the AR of magnetite to SA106 Gr.B.

## Figures and Tables

**Figure 1 materials-12-00628-f001:**
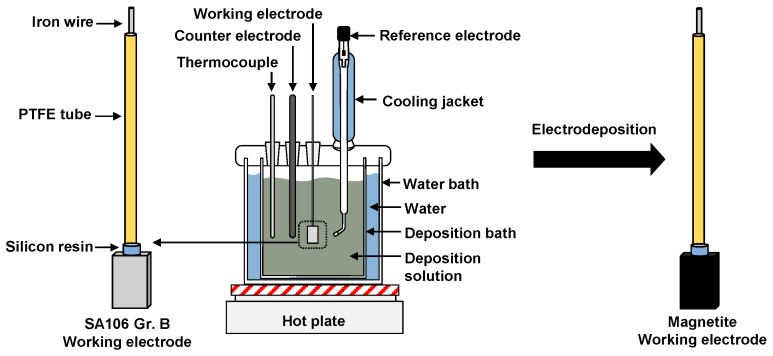
Schematic of the test apparatus for the preparation of SA106 Gr.B and magnetite working electrodes. Polytetrafluoroethylene (PTFE) tube

**Figure 2 materials-12-00628-f002:**
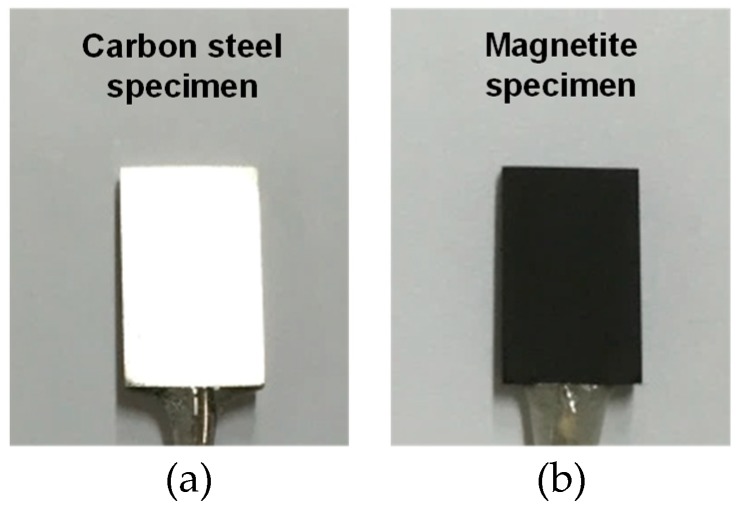
Photographs of SA106 Gr.B and electrodeposited magnetite specimens: (**a**) SA106 Gr.B specimen and (**b**) magnetite specimen.

**Figure 3 materials-12-00628-f003:**
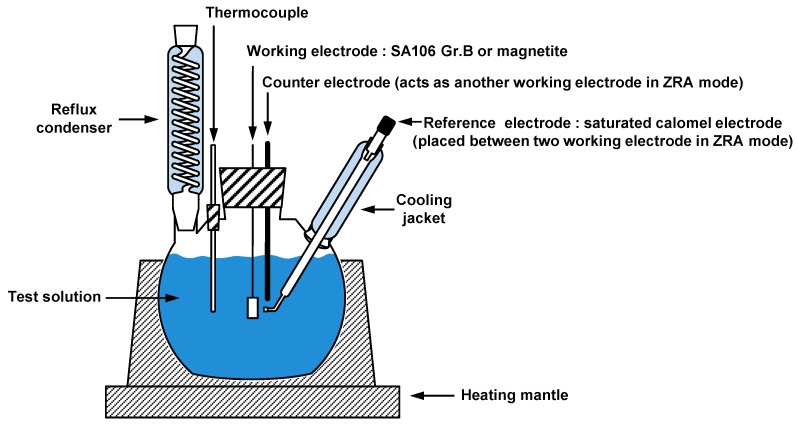
Schematic of the three-electrode cell for the electrochemical corrosion tests. Zero resistance ammeter (ZRA).

**Figure 4 materials-12-00628-f004:**
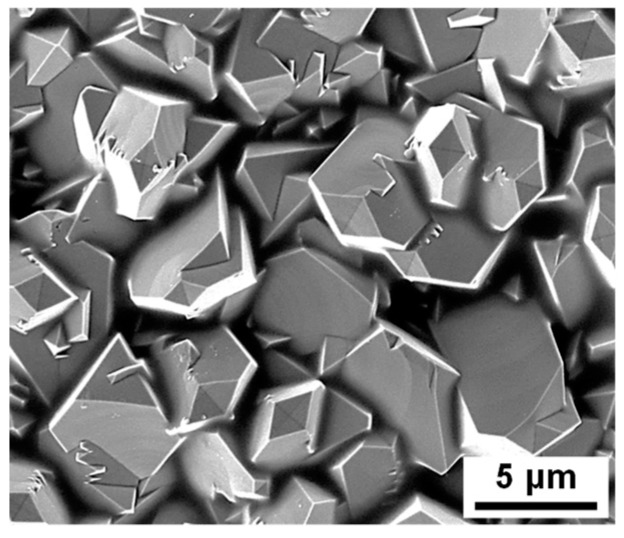
Scanning electron microscope (SEM) micrograph of surface of the magnetite layer electrodeposited on the SA106 Gr.B substrate.

**Figure 5 materials-12-00628-f005:**
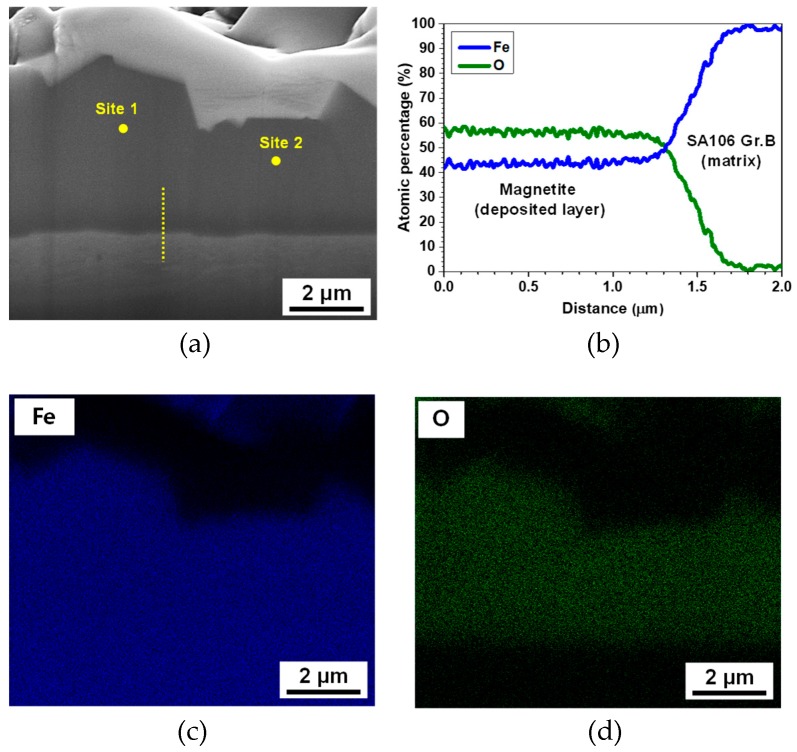
Focused ion beam-scanning electron microscopy (FIB-SEM) micrographs and energy dispersive X-ray spectroscope (EDS) mapping analysis of the magnetite layer electrodeposited on the SA106 Gr.B substrate: (**a**) cross-section of magnetite, (**b**) EDS line analysis, (**c**) Fe element, and (**d**) O element of the EDS mapping results.

**Figure 6 materials-12-00628-f006:**
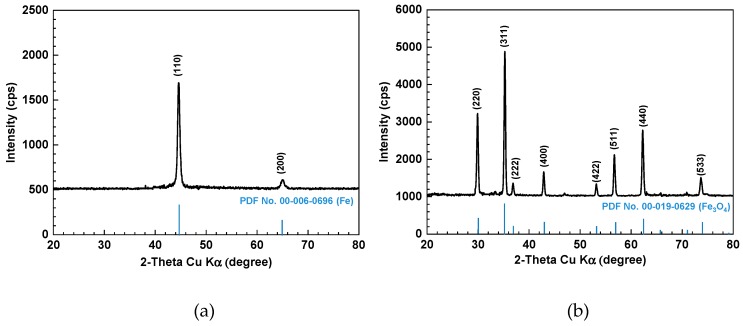
Results of the X-ray diffractometer (XRD) analysis of the SA106 Gr.B and electrodeposited magnetite layer on the SA106 Gr.B substrate: (**a**) SA106 Gr. B and (**b**) magnetite.

**Figure 7 materials-12-00628-f007:**
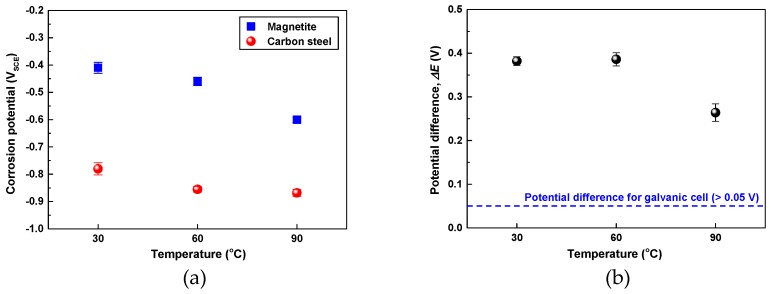
Effect of temperature on the corrosion potentials of SA106 Gr.B and magnetite and potential differences between SA106 Gr.B and magnetite in an alkaline solution at 30, 60, and 90 °C: (**a**) corrosion potentials and (**b**) the differences in corrosion potentials between SA106 Gr.B and magnetite.

**Figure 8 materials-12-00628-f008:**
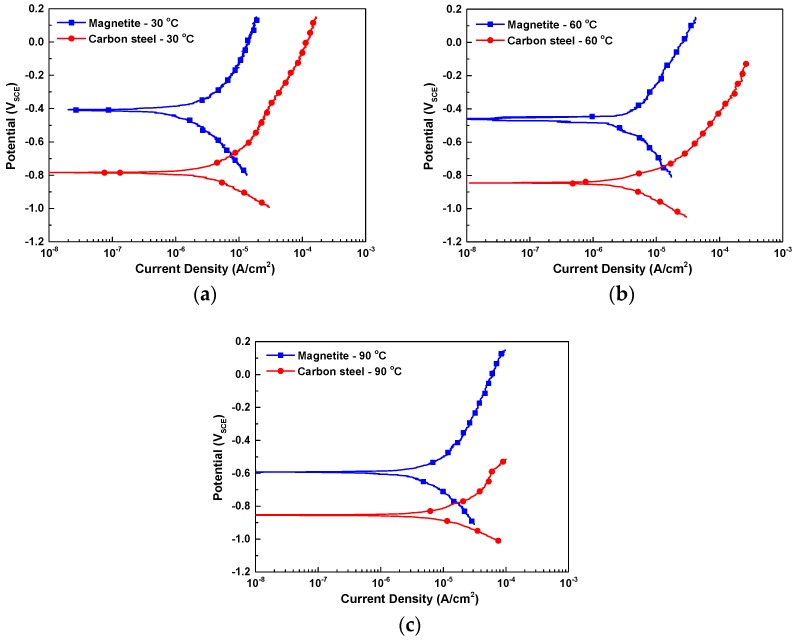
Results of potentiodynamic polarization test of SA106 Gr.B and magnetite in an alkaline solution at various temperature conditions: (**a**) 30 °C, (**b**) 60 °C, and (**c**) 90 °C.

**Figure 9 materials-12-00628-f009:**
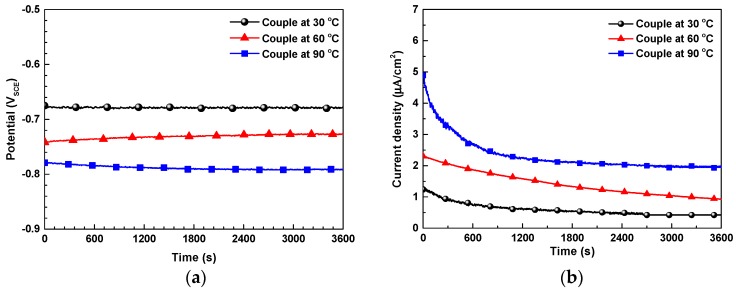
Galvanic parameters of SA106 Gr.B coupled to magnetite measured from the ZRA measurements in test solutions at various temperatures: (**a**) galvanic potential and (**b**) galvanic current density.

**Figure 10 materials-12-00628-f010:**
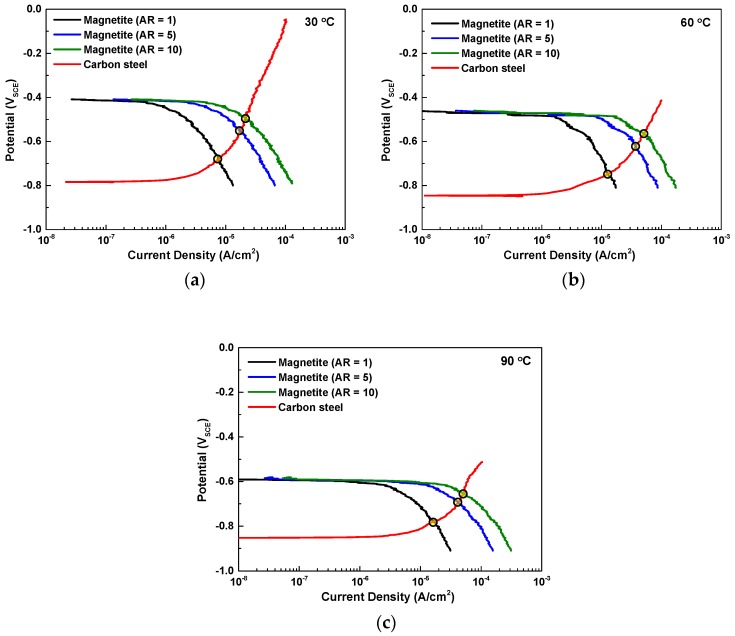
Effect of the AR on the galvanic behavior between SA106 Gr.B (anodic curves) and magnetite (cathodic curves) in an alkaline solution at various temperatures: (**a**) 30 °C, (**b**) 60 °C, and (**c**) 90 °C.

**Table 1 materials-12-00628-t001:** Chemical composition of SA106 Gr.B carbon-manganese steel (wt. %).

Ni	Cr	Mo	Si	Mn	Cu	C	P	S	Fe
0.2	0.4	0.1	0.23	1.05	0.1	0.19	0.012	0.005	Bal.

**Table 2 materials-12-00628-t002:** Point EDS analysis of the magnetite layer electrodeposited on the SA106 Gr.B substrate.

Element (at.%)	Site 1	Site 2
Fe	43.4	42.8
O	56.6	57.2

**Table 3 materials-12-00628-t003:** Various electrochemical parameters of SA106 Gr.B and magnetite via cathodic Tafel extrapolation and mixed potential theory (area ratio (AR) = 1).

Temp.	Materials	Electrochemical Parameters		Galvanic Parameters	
		*E*_corr_ (V_SCE_)	*i*_corr_ (μA/cm^2^)	*E*_couple_ (V_SCE_)	*i*_couple_ (μA/cm^2^)
30 °C	SA106 Gr.B	−0.783 ± 0.004	2.51 ± 0.02	−0.682 ± 0.024	7.36 ± 0.21
	Magnetite	−0.407 ± 0.005	0.69 ± 0.03		
60 °C	SA106 Gr.B	−0.841 ± 0.006	3.42 ± 0.12	−0.746 ± 0.023	12.42 ± 0.32
	Magnetite	−0.455 ± 0.005	1.24 ± 0.12		
90 °C	SA106 Gr.B	−0.852 ± 0.010	6.91 ± 0.18	−0.778 ± 0.021	17.13 ± 0.37
	Magnetite	−0.592 ± 0.008	2.44 ± 0.18		

**Table 4 materials-12-00628-t004:** Galvanic current density *i*_couple_ at various temperature and AR = 1, 5, and 10 obtained via mixed potential theory.

Temp.	Galvanic Current Density (μA/cm^2^)		
	AR = 1	AR = 5	AR = 10
30 °C	7.36 ± 0.21	17.43 ± 1.20	22.23 ± 2.25
60 °C	12.42 ± 0.32	37.12 ± 1.58	52.05 ± 3.34
90 °C	17.13 ± 0.37	41.43 ± 1.74	51.82 ± 3.51

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
