# Peer review of "Galvanic Corrosion of SA106 Gr.B Coupled with Magnetite in Alkaline Solution at Various Temperatures"

_materials, 2019, doi:10.3390/ma12040628_

Round 1

Reviewer 1 Report

(1) Used instruments should be clearly addressed in the experimental methods part.

(2) Typo on ZRA, line 117.

(3) X-axis in XRD analysis should be mentioned either in copper (Cu) or cobalt (Co).

Author Response

Dear Editor,

I have revised and resubmitted the manuscript entitled “Galvanic Corrosion of SA106 Gr.B Coupled with Magnetite in Alkaline Solution at Various Temperature”. (Manuscript No.: materials-434556).

Thanks to the reviewers’ kind reviews and comments, we had a valuable chance to improve the quality of our manuscript. We have thoroughly revised our manuscript to accommodate all the comments given by the reviewers, although some of them seem to be a little bit insufficient.

The attached letter contains our detailed responses to the reviewers’ comments. The revised parts of the manuscript are highlighted in yellow (reviewer 1), blue (reviewer 2), and green (reviewer 3) so that you can easily view the changes. Please find the author’s response to the reviewers’ comments on the next page, where we summarized our answers to each of the comments. In the process of revising the paper, Dr. Sang Ji Kim performed the EDS and XRD analysis and reviewed the manuscript. Hence, we newly added the Dr. Sang Ji Kim as a co-author in this study.

I Sincerely hope that the reviewer's comments are properly accommodated in the reivsed manuscript so that it is accetable for publication in Materials.

Happy New Year!  

Yours sincerely,

Do Haeng Hur

Reviewer 2 Report

The manuscript "Galvanic Corrosion of SA106 Gr.B Coupled with Magnetite in Alkaline Solution of Different temperature” by Soon-Hyeok Jeon et al. reports the effect of three different temperatures on the galvanic corrosion behavior of SA106 Gr.B carbon steel coupled with magnetite. The manuscript is well structured with clear messages and nice readable graphs. The results are helpful to other researchers in this field. However, there are various points that must be re-considered before publication in any case:

In introduction section, the authors should be addressed recent papers and highlight your hypothesis, new concepts and innovations briefly. In addition, authors should introduce SA106 Gr.B as a carbon-manganese steel and highlight the applications.

The corrosion behavior of SA106 Gr.B Coupled with Magnetite in Alkaline Solution was already studied on other report by Soon-Hyeok Jeon et al. (Materials Transactions, Vol. 56, No. 7 (2015) pp. 1107 to 1111), so the novelty of this work seems to be rather poor. Authors must introduce this reference in the present manuscript and demonstrate the novelty of this work.

Have the authors done the electrochemical study of the magnetite bath on SA106 Gr.B? I suggest cyclic voltammetry analyses for this investigation. The authors should give more details on the selection of the selected electrodeposition potential.

Further characterization such as EDS line analysis on the cross-section of magnetite (two points are insufficient) and XRD analysis of SA106 Gr.B… can be provided to improve the quality of the work.

Author Response

Dear Editor,

I have revised and resubmitted the manuscript entitled “Galvanic Corrosion of SA106 Gr.B Coupled with Magnetite in Alkaline Solution at Various Temperature”. (Manuscript No.: materials-434556).

Thanks to the reviewers’ kind reviews and comments, we had a valuable chance to improve the quality of our manuscript. We have thoroughly revised our manuscript to accommodate all the comments given by the reviewers, although some of them seem to be a little bit insufficient.

The attached letter contains our detailed responses to the reviewers’ comments. The revised parts of the manuscript are highlighted in yellow (reviewer 1), blue (reviewer 2), and green (reviewer 3) so that you can easily view the changes. Please find the author’s response to the reviewers’ comments on the next page, where we summarized our answers to each of the comments. In the process of revising the paper, Dr. Sang Ji Kim performed the EDS and XRD analysis and reviewed the manuscript. Hence, we newly added the Dr. Sang Ji Kim as a co-author in this study.

I sincerely hope that the reviewer's comments are properly accommodated in the reivsed manuscript so that it is accetable for publication in Materials.

Happy New Year! 

Yours sincerely,

Do Haeng Hur

Reviewer 3 Report

This work deals with "Galvanic Corrosion of SA106 Gr.B Coupled with  Magnetite in Alkaline Solution of Different temperature".  This problem is interesting  for industrial applications and the authors give a good presentations in their study. However, there are questions which should be improved before accepting. 

The title of Galvanic Corrosion of SA106 Gr.B Coupled with  Magnetite in Alkaline Solution of Different temperature at line 2 is not correct, because the "Different temperature' should be plural. 

At line 12, as shown as At all temperatures studied, carbon steel acted as the of the galvanic cell composed of carbon steel and magnetite because the corrosion potential of carbon steel was significantly lower than that of magnetite. Here "the of " in carbon steel acted as the of the galvanic cell seems to be deleted.

In section of 2.1. Specimen Preparation and Test Solution, the authors should indicate the vacuum degree which in important to understand the oxidation.

In section of 2.1. Specimen Preparation and Test Solution, it is also important to know the operation parameters for FIB-SEM, such as working distance and kV.

In Figure 2 there are two photos, which should be labelled as (a) and (b).

The discussions of Figure 7 seems to be before the discussions of Figure 6, because Figure 7 is raw material.

Author Response

(The authors gave the same response as above.)

Round 2

Reviewer 2 Report

I recommend to accept the revised manuscript.